# African eggplant-associated virus: Characterization of a novel tobamovirus identified from *Solanum macrocarpon* and assessment of its potential impact on tomato and pepper crops

Anne K. J. Giesbers[1,2]*, Annelien Roenhorst[1,2], Martijn F. Schenk[2], Marcel Westenberg[1,2], Marleen Botermans[1,2]

1 Netherlands Institute for Vectors, Invasive plants and Plant health, Wageningen, The Netherlands,
2 National Plant Protection Organization (NPPO-NL), Netherlands Food and Consumer Product Safety Authority (NVWA), Wageningen, The Netherlands

* a.k.j.giesbers@nvwa.nl

**Data Availability Statement:** Sequence data are available from the GenBank database: accession

## Abstract

A novel tobamovirus was identified in a fruit of *Solanum macrocarpon* imported into the Netherlands in 2018. This virus was further characterized in terms of host range, pathotype and genomic properties, because many tobamoviruses have the potential to cause severe damage in important crops. In the original fruit, two different genotypes of the novel virus were present. The virus was able to infect multiple plant species from the *Solanaceae* family after mechanical inoculation, as well as a member of the *Apiaceae* family. These species included economically important crops such as tomato and pepper, as well as eggplant and petunia. Both tomato and pepper germplasm were shown to harbor resistance against the novel virus. Since most commercial tomato and pepper varieties grown in European greenhouses harbor these relevant resistances, the risk of infection and subsequent impact on these crops is likely to be low in Europe. Assessment of the potential threat to eggplant, petunia, and other susceptible species needs further work. In conclusion, this study provides a first assessment of the potential phytosanitary risks of a newly discovered tobamovirus, which was tentatively named African eggplant-associated virus.

## Introduction

The International Plant Protection Convention (IPPC) aims to safeguard agriculture and natural resources against the entry, establishment and spread of economically and environmentally significant pests [1]. In this respect, import inspections are an important tool to prevent the introduction of regulated plant pests and may serve as an early warning system on emerging pests. In 2018, a phytosanitary inspector in the Netherlands intercepted a fruit of *Solanum macrocarpon* from Mexico with symptoms that indicated a potential viral infection. *Solanum macrocarpon*, also known as African eggplant, is a small tropical perennial originating from

numbers OP729507-OP729508. All other relevant data are within the manuscript and its Supporting Information files.

**Funding:** The author(s) received no specific funding for this work.

**Competing interests:** The authors have declared that no competing interests exist.

Africa that is nowadays cultivated in many countries for food, medicinal purposes and as an ornamental [2, 3].

Samples of the *S. macrocarpon* fruit were analyzed by bioassay, serology and high-through-put sequencing analysis. The results indicated the presence of a so far unknown tobamovirus. The genus *Tobamovirus* (family *Virgaviridae*) currently contains 37 accepted species according to the 2021 release of the International Committee on Taxonomy of Viruses (ICTV) (https://talk.ictvonline.org/taxonomy/), including some devastating plant viruses. Tobamoviruses are highly stable, contagious and difficult to control. Tobamoviruses can rapidly spread locally by mechanical contact, while transmission via seeds enables them to spread over longer distances, as exemplified by the recent rapid spread of tomato brown rugose fruit virus (ToBRFV) [4–6] and tomato mottle mosaic virus (ToMMV) [7]. Other tobamoviruses that can cause significant economic losses in crops if not controlled effectively, include cucumber green mottle mosaic virus [8], tobacco mosaic virus (TMV) and tomato mosaic virus (ToMV) [9].

The potential of tobamoviruses to spread rapidly and cause severe damage has prompted us to prioritize this novel virus for further characterization in terms of genomic properties, host range and pathotype. Tobamoviruses can be divided into at least eight subgroups based on their complete nucleotide sequence, the amino acid sequences of each individual viral protein and their host plant family [10]. These subgroups consist of species that can infect plant species within the families *Apocynaceae*, *Brassicaceae*, *Cactaceae*, *Cucurbitaceae*, *Leguminosae*, *Malvaceae*, *Passifloraceae*, and *Solanaceae*.

To assess the biological properties and potential impact of the novel tobamovirus from *S. macrocarpon*, host range studies focused on solanaceous crops, in particular tomato (*Solanum lycopersicum*) and pepper (*Capsicum* spp.). Both tomato and pepper germplasm are known to harbor tobamovirus resistance genes. In tomato, the *Tm-1* gene, and the *Tm-2* and *Tm2*$^2$ alleles, originating from wild tomato species, have been used extensively to protect against tobacco mosaic virus and tomato mosaic virus [11]. In pepper, the four *L* genes or alleles $L^1$, $L^2$, $L^3$ and $L^4$, each provide resistance to an increasing number of tobamovirus pathotypes [12]. Therefore, tomato and pepper accessions harboring these resistances were tested to assess whether the novel tobamovirus poses a threat to these crops.

The purpose of the molecular and biological characterization of this novel tobamovirus from *S. macrocarpon* was to contribute to the assessment of its potential risks, so that phytosanitary authorities can determine the need for preventive control measures.

## Materials and methods

### Virus propagation, maintenance and mechanical inoculation

A novel tobamovirus was identified from a fruit of *S. macrocarpon* imported from Mexico. The isolate was propagated and maintained by successive mechanical passages in *Nicotiana occidentalis* P1. Symptomatic tissue (fruit or leaf material) was used for mechanical inoculation of test plants according to the procedure and conditions as described by Verhoeven and Roenhorst (2000) [13]. Young leaves were harvested just after systemic symptoms appeared and stored at -80˚C for further experiments as well as long-term preservation.

### Host range and pathotype analysis

Nineteen species from six plant families were inoculated with the virus from *S. macrocarpon*. Plants were grown from seed, except for *Solanum tuberosum* which was grown from tubers. Tested plant species and varieties are specified in the Results section. Per species, four plants were inoculated with the virus and four plants with inoculation buffer only (mock treatment). All plants were assessed for local and systemic disease symptoms twice a week for a total

duration of four weeks and tested by DAS-ELISA 10–25 days after inoculation. Symptoms in *S. macrocarpon* were assessed until the fruits had matured. In addition, tomato and pepper accessions with different tobamovirus resistance genes (tomato: *Tm1*, *Tm2* and *Tm2²*; pepper: $L^1$, $L^2$, $L^3$ and $L^4$) were tested in an experiment with the same set-up. These accessions and information on their resistance genes were kindly provided by the Centre for Genetic Resources, the Netherlands (CGN).

## Double Antibody Sandwich Enzyme-Linked Immunosorbent Assay (DAS-ELISA)

DAS-ELISA [14] was performed using polyclonal antisera raised against bell pepper mottle virus (BPMoV), pepper mild mottle virus (PMMoV), tobacco mosaic virus (TMV), and tomato mosaic virus (ToMV-D) from Prime Diagnostics, the Netherlands. Host range and pathotype experiments were conducted with PMMoV antiserum. Antisera were diluted at 1:1000 from a 1 mg/ml stock. Buffers used were: coating buffer (15 mM $Na_2CO_3$, 34.9 mM $NaHCO_3$, pH 9.6), homogenization buffer (2.9 mM $KH_2PO_4$, 2.7 mM KCl, 16 mM $Na_2H$-$PO_4 \cdot 12H_2O$, 136.9 mM NaCl, containing Tween 20 at 0.5 ml/l, polyvinyl pyrrolidone at 20 g/l and albumin from bovine serum at 2.0 g/l, pH 7.4), and substrate buffer (p-nitrophenyl phosphate at 0.75 mg/ml in diethanolamine at 97.5 ml/l, pH 9.8). A volume of 200 μl was added to each well. Each sample was tested in duplicate and contained a pool of tissue from four plants of the same combination of plant species and inoculum. Samples were taken from non-inoculated young leaves or fruits and homogenized at an approximate ratio of 1:10 (w:v), using extraction bags and a hand-held homogenizer (Bioreba). Coating and test sample incubations were overnight at 4°C, substrate incubations for 1–2 hours at 37°C. Optical Density (OD) at 405 nm was measured with a microplate reader (BioRad iMark) in duplicate wells from which an average OD value was calculated. Samples were considered positive when the OD value was at least twice the value of the negative controls (mock-inoculated healthy plants).

## Illumina sequencing and analysis

Total RNA was extracted using the RNeasy Plant Mini kit (Qiagen, Germany) following the manufacturer's instructions. The RNA extract was sent to GenomeScan (The Netherlands) for generation of 2 Gb of Illumina RNAseq 150PE (paired-end) data per sample. Sequencing was performed on an Illumina NovaSeq6000. RNAseq data were analyzed in CLC Genomics workbench v21.0.4 (Qiagen, Germany) and run in a custom workflow built for detection of *de novo* assembled viral contigs [15]. Consensus sequences (>100 nt; ARC>10) from *de novo* assembled contigs were analyzed using megaBLAST and DIAMOND [16] with a local installation of the NCBI nr/nt database downloaded from http://ftp.ncbi.nlm.nih.gov/. BLAST results were visualized in Krona (bitscore threshold = 25) [17]. Putative viral sequences were further analyzed using Geneious Prime 2021.1.1 (Biomatters, New Zealand). Genomes of *Tobamovirus* species were aligned using MAFFT alignment with automatic settings in Geneious, for the species of which a complete reference nucleotide sequence was available in GenBank. A neighbor joining tree was constructed using the Geneious tree builder based on Tamura-Nei distances, bootstrap resampling, random seed 925,045, number of replicates 1,000, and a support threshold at 70%. In addition, a maximum likelihood tree was run using Geneious plugin RAxML 4.0 with 1,000 replicates and all other variables set as default.

## Results

In 2018, a phytosanitary inspector in the Netherlands intercepted a fruit of *Solanum macrocarpon* from Mexico, which showed diffuse chlorotic mottling (Fig 1). To determine whether the

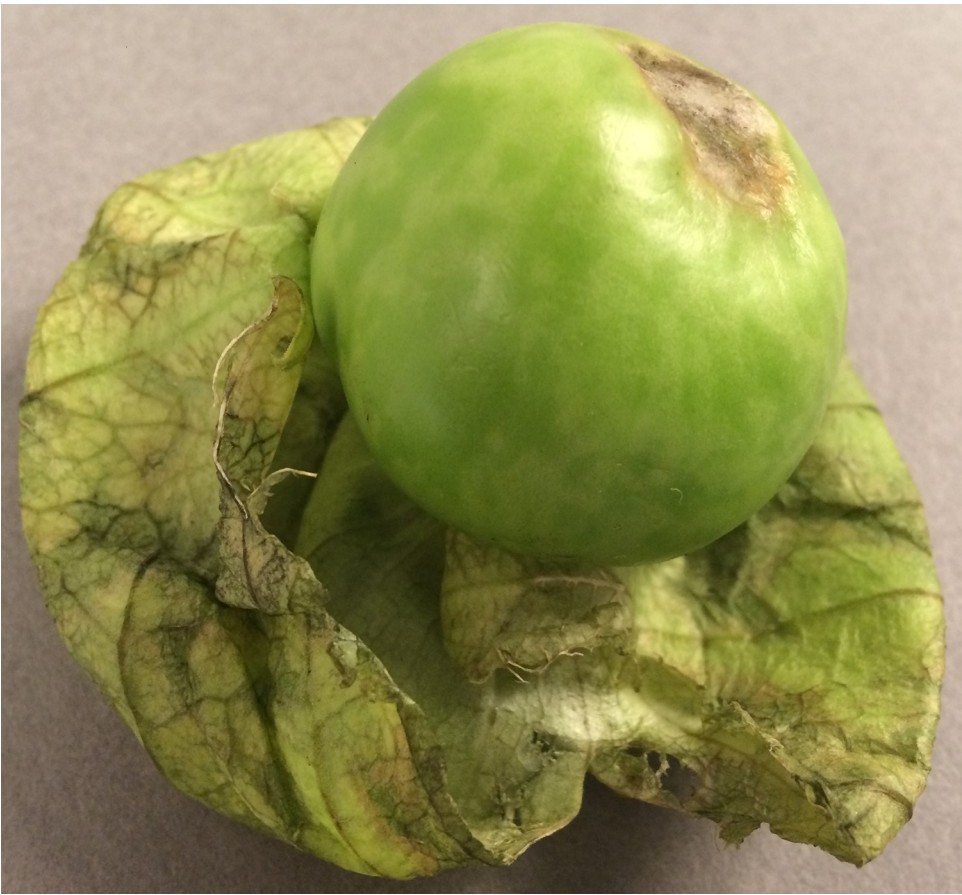

**Fig 1. *Solanum macrocarpon* fruit with diffuse chlorotic mottling.**

symptoms on the *S. macrocarpon* fruit could result from a viral infection, a homogenate of symptomatic tissue was inoculated on a range of test plants routinely used as indicators in virus diagnosis. The reaction on the inoculated test plants, six days post inoculation, indicated the presence of a tobamovirus (Table 1). Therefore, symptomatic leaves of *N. occidentalis* P1 were tested for different tobamoviruses by DAS-ELISA. Reactions were clearly positive for

**Table 1. Symptomatology of indicator plants inoculated with the homogenate from the intercepted *S. macrocarpon* fruit.**

| Indicator species | Symptoms | |
|---|---|---|
| | Local | Systemic |
| *Datura stramonium* | NL (2) | - |
| *Nicotiana glutinosa* | NL (2) | - |
| *N. occicentalis* "P1" | - | ~CV, G, LN (2) |
| *N. tabacum* "White Burley" | CL, NL (2) | - |
| *Solanum lycopersicum* "Moneymaker" | - | - |

-: no symptoms, ~: mild symptoms, CV: chlorotic vein banding, CL: chlorotic lesions, G: growth inhibition, LN: leaf narrowing, NL: necrotic lesions. The first symptoms appeared six days post inoculation. Per species two plants were inoculated. The number of plants showing specific symptoms is indicated in brackets.

**Table 2. Reaction of inoculated *N. occidentalis* P1 plants by DAS-ELISA.**

| Antiserum | OD$_{405}$* |
|---|---|
| BPMoV | 3.050 |
| PMMoV | 1.799 |
| TMV | 0.228 |
| ToMV-D | 0.269 |

* Average OD values after 90 minutes of substrate incubation. OD values were considered positive (red) when ≥ 2x the OD value of mock-inoculated controls, otherwise OD values were considered negative (green).

antisera raised against BPMoV and PMMoV, but negative for TMV and ToMV-D (Table 2). Negative controls (mock-inoculated plants) and homologous (positive) controls showed their expected outcomes. Based on these results, the virus from *S. macrocarpon* appeared to be a tobamovirus that is serologically related to BPMoV and PMMoV. The PMMoV antiserum was used to detect the virus in all subsequent experiments.

For further characterization, the viral genome sequence of the putative tobamovirus from *S. macrocarpon* was determined by Illumina sequencing analysis of the original fruit sample. *De novo* assembly resulted in a near-complete genome sequence of two genotypes, each consisting of a single stranded RNA with four predicted open readings frames (ORFs), corresponding to the genome organization of tobamoviruses [18]. These respective ORFs encode a small replicase subunit, an RNA-dependent RNA polymerase (translated through ribosomal readthrough at the stop codon of the small replicase subunit), a movement protein, and a coat protein. Genotype 1 (GenBank accession number OP729507) and genotype 2 (GenBank accession number OP729508) contained 6228 and 6356 nucleotides respectively, with 96.1% nucleotide similarity. No other viruses were detected.

NCBI BLAST searches (https://blast.ncbi.nlm.nih.gov/Blast.cgi) indicated that both genotypes of the virus showed most resemblance to isolates of PMMoV, with a maximum query coverage of 97% and a nucleotide identity of 75%. Since the nucleotide identity with PMMoV is considerably lower than the tobamovirus species demarcation threshold of 90% [18], the virus should be considered as a member of a new species, for which we propose the name "African eggplant-associated virus" (AEaV). A phylogenetic analysis including genome sequences of other tobamoviruses indicated that AEaV is most closely related to members of species infecting solanaceous plants, in particular PMMoV and tropical soda apple mosaic virus (Fig 2). The relative positioning of each virus within the tree was the same for the neighbor joining as for the maximum likelihood (RAxML) method.

BLAST searches also indicated a close resemblance to a bacteriophage isolate (GenBank accession number BK018885), which aligned with the first 3,874 nucleotides of the novel virus, corresponding to a query coverage of 60%. The nucleotide identity was 95.7% and 99.5% for genotype 1 and genotype 2, respectively. Most probably, this sequence was incorrectly assigned to a bacteriophage.

To gain more insight into the potential host range of the novel virus, it was mechanically transferred from *N. occidentalis* P1 to test plants (Table 3). Nine out of 19 tested species, including *S. macrocarpon*, were susceptible based on symptomatology (i.e. systemic symptoms) and/or detection by DAS-ELISA, of which *Ammi majus* was the only non-solanaceous plant. *A. majus*, *S. lycopersicum* "Moneymaker" and *Solanum melongena* "Violetta Lunga 2" showed asymptomatic or latent infections. The tested *Amaranthaceae*, *Brassicaceae*, *Cucurtitaceae*, and *Fabaceae* species remained uninfected. Negative and positive controls showed their expected outcomes. Fig 3 illustrates the most obvious symptoms per species.

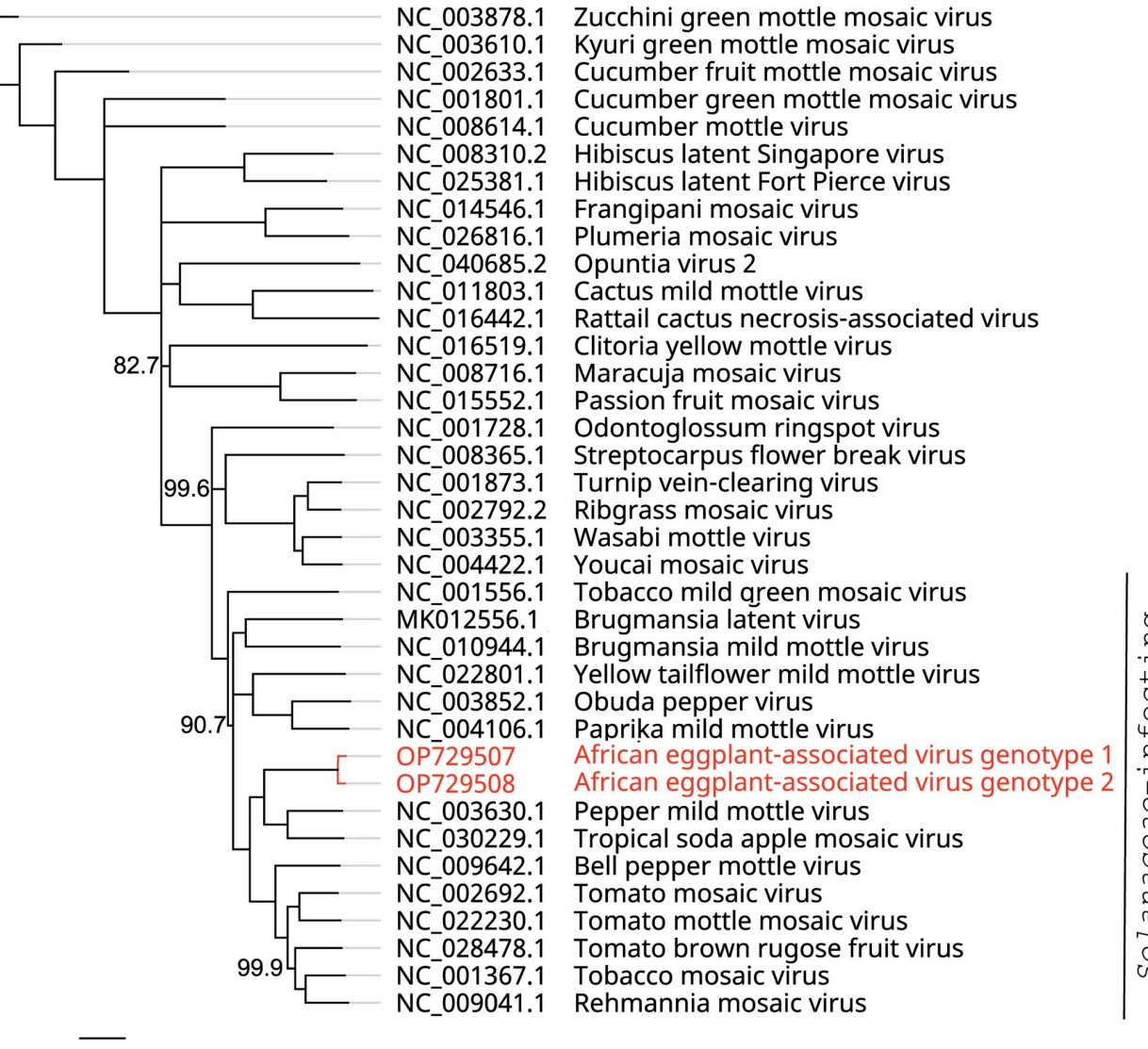

**Fig 2. Unrooted neighbor-joining tree with (near) complete nucleotide sequences of the novel virus (red) and other members of the genus** *Tobamovirus* **including their GenBank accession numbers.** The bar represents 0.1 substitutions per site. This tree was constructed by the neighbor-joining algorithm in Geneious software, using the genetic distance model Tamura-Nei, with 1,000 bootstrap replicates and the support threshold set at 70%. Support values (%) that were less than 100% are indicated next to the supported internal branches.

To fulfill Koch's postulates, *S. macrocarpon* plants were observed until fruits had matured, 147 days after inoculation. The initial virus symptoms on the leaves had disappeared completely ten days after their first observation. For mature plants and fruits, the virus was still detected by DAS-ELISA, but there were no phenotypic differences compared to mock-inoculated plants and fruits. Therefore, associating the novel virus with the symptoms observed on the original fruit (Fig 1), according to Koch's postulates, was not possible.

Furthermore, in the inoculated *S. macrocarpon* plants only genotype 2 was detected by Illumina sequence analysis. To trace the disappearance of genotype 1, subsequent passages in *N. occidentalis* P1 used for virus propagation were analyzed. Both genotypes were still detected in all passages and their nucleotide sequence remained exactly the same, though the relative

**Table 3. Host range study: Symptomatology and DAS-ELISA results of test plants after inoculation with the novel tobamovirus.**

| Family | Species | symptoms | | OD$_{405}$ |
|---|---|---|---|---|
| | | local | systemic | PMMoV |
| *Amaranthaceae* | *Chenopodium album* | CL (4) | - | 0.022 |
| | *Chenopodium quinoa* | CL (4), ~CR (2) | - | 0.135 |
| *Apiaceae* | *Ammi majus* | - / -* | -; -* | 1.169 / 0.681* |
| *Brassicaceae* | *Brassica rapa* subsp. *pekinensis* "Granaat" | - | - | 0.105 |
| *Cucurbitaceae* | *Cucumis sativus* "Chinese Slangen" | - | - | 0.000 |
| *Fabaceae* | *Pisum sativum* "Kelvedon Wonder" | - | - | 0.052 |
| | *Vicia faba* "Witkiem" | - | - | 0.071 |
| *Solanaceae* | *Capsicum annuum* "Westlandse Grote Zoete" | - | C, M, Ms (4) | 1.353 |
| | *Datura stramonium* | NL (4) | - | 0.149 |
| | *Nicotiana glutinosa* | NL (4) | - | 0.048 |
| | *Nicotiana hesperis* "67A" | - | C, G, NLE, R (4) | 0.768 |
| | *Nicotiana occidentalis* "P1" | - | C, G, LC, NLE, R (4) | 1.653 |
| | *Nicotiana tabacum* "White Burley" | CL (4), NL (1) | - | 0.101 |
| | *Petunia hybrida* | NL, NV (1) | LC (3) G, M, R (4) | 0.989 |
| | *Solanum lycopersicum* "Moneymaker" | - | - | 0.631 |
| | *Solanum macrocarpon* | - | CV, ~M (2) | 2.340 |
| | *Solanum melongena* "Violetta Lunga 2" | - / -* | - / -* | 0.763 / 1.788* |
| | *Solanum nigrum* | - | CL (3) R (2) | 0.647 |
| | *Solanum tuberosum* "Nicola" | - | - | 0.090 |

Per species four plants were inoculated; the number of plants showing specific symptoms is indicated in brackets. Average OD values after two hours of substrate incubation were considered positive (red) when $\geq$ 2x the value of mock-inoculated controls, otherwise values were considered negative (green). -: no symptoms, ~: mild symptoms, C: chlorosis, CL: chlorotic lesions, CR: chlorotic rings, CV: chlorotic vein banding, G: growth inhibition, LC: leaf curling, M: mottling, Ms: mosaic, NL: necrotic lesions, NLE: necrotic leaf edge, NV: necrotic vein banding, R: rugosity

* results of a repeated experiment on four new plants.

frequency of genotype 1 decreased with each passage. Based on sequence depth, the approximate ratios of genotype 1 to genotype 2 were 2:3 for the original *S. macrocarpon* fruit, 1:4 for *N. occidentalis* P1 (passage 1), and 1:14 for *N. occidentalis* P1 (passage 2) (S1 Fig).

To assess whether the novel tobamovirus poses a threat to tomato and pepper crops, accessions harboring different tobamovirus resistances were inoculated (Table 4). These included (combinations of) *Tm1*, *Tm2*, and *Tm2²*, and *L¹*, *L²*, *L³* and *L⁴* resistance genes/alleles for tomato and pepper, respectively. Even though all tomato accessions remained asymptomatic, only the four accessions harboring the combination of *Tm1* and *Tm2*/*Tm2²* tested negative by DAS-ELISA. Interestingly, the accessions that harbored *Tm1*, *Tm2* or *Tm2²* separately tested positive by DAS-ELISA. Therefore, a combination of these *Tm* resistance genes seems to account for resistance against the novel tobamovirus.

In contrast to the previous experiment (Table 3) in which all plants of the pepper cultivar "Westlandse Grote Zoete" showed systemic symptoms, only two out of four inoculated plants showed local symptoms, whereas the other two plants showed systemic symptoms only. However, since cultivar "Westlandse Grote Zoete" does not harbor any tobamovirus resistance genes the presence of the virus was expected in all plants, which was indeed confirmed by DAS-ELISA (Table 4). Furthermore, all pepper accessions harboring a tobamovirus resistance gene/allele showed local but no systemic symptoms (S2 Fig), and tested negative by DAS-ELISA (Table 4). Therefore, it appears that each individual *L¹*, *L²*, *L³* and *L⁴* gene/allele provided

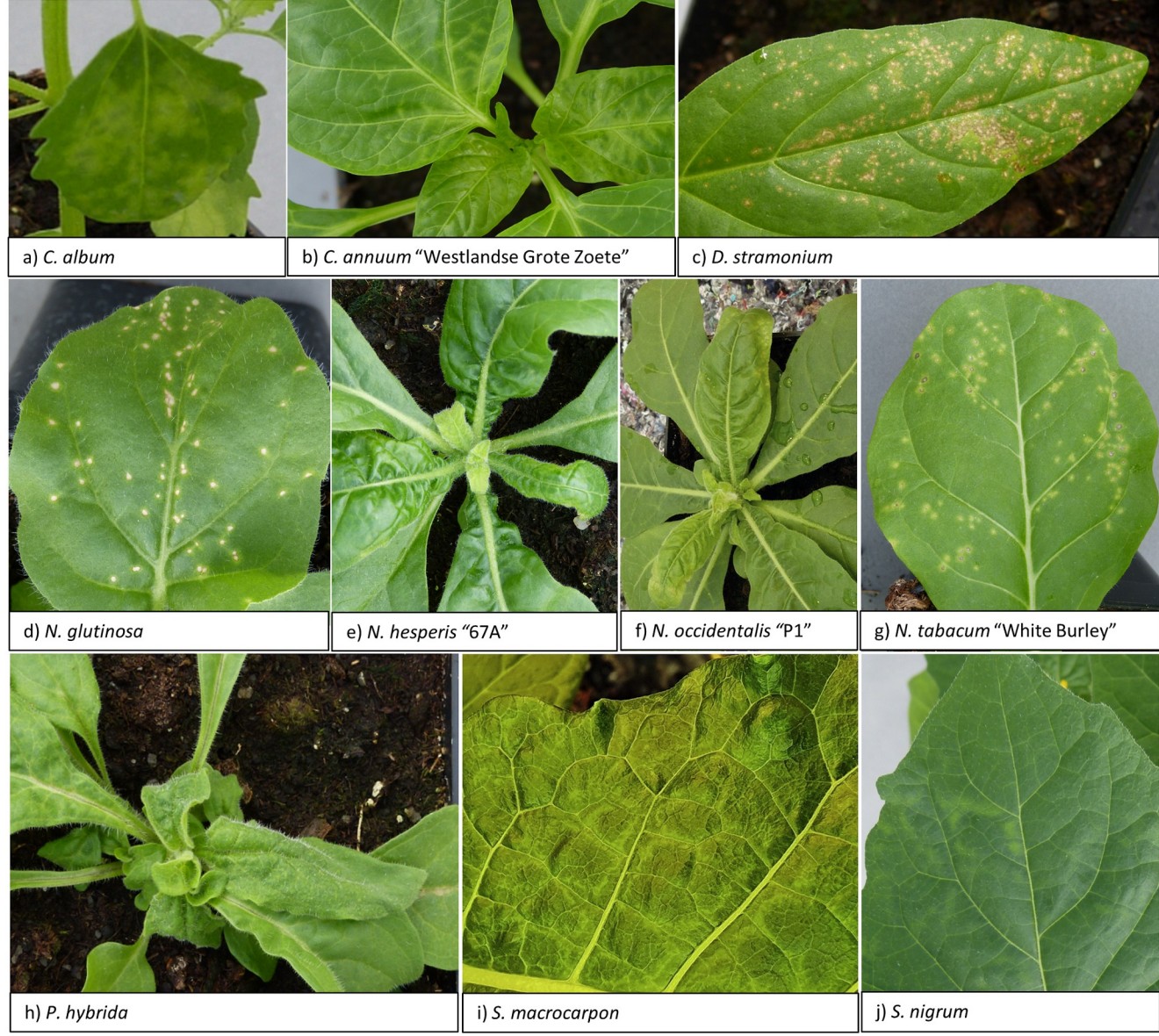

**Fig 3. Symptoms on test plants upon inoculation with the novel tobamovirus.** a) *Chenopodium album*: local chlorotic lesions, b) *Capsicum annuum* "Westlandse Grote Zoete": systemic chlorosis, c) *Datura stramonium*: local necrotic lesions, d) *Nicotiona glutinosa*: local necrotic lesions, e) *Nicotiana hesperis* "67A": systemic chlorosis, growth inhibition, rugosity, f) *Nicotiana occidentalis* "P1": systemic chlorosis, growth inhibition, leaf curling, rugosity, g) *Nicotiana tabacum* "White Burley": local chlorotic lesions, h) *Petunia hybrida*: local necrotic lesions, systemic leaf curling, growth inhibition, rugosity, i) *Solanum macrocarpon*: systemic chlorotic vein banding, j) *Solanum nigrum*: systemic chlorotic lesions, rugosity.

resistance to the novel tobamovirus. Overall, these results showed that resistance against the novel virus is present in both the tomato and pepper germplasm.

Finally, to gain more insight into the geographic distribution and the host range of the novel virus, both symptomatic and asymptomatic fruits of *S. macrocarpon* and other *Solanaceae* imported into the Netherlands in 2021, were subjected to Illumina sequencing analysis. This analysis included 19 samples from four countries, but no further material from Mexico: *Solanum betaceum* from Colombia (2 samples), *S. macrocarpon* from Colombia (1 sample), *S. macrocarpon* from Surinam (10 samples), *S. melongena* from Malaysia (1 sample), and

**Table 4. Pathotype analysis: Symptomatology and DAS-ELISA results of tomato (*S. lycopersicum*) and pepper (*Capsicum* spp.) accessions harboring different (combinations of) tobamovirus resistance genes after inoculation with the novel tobamovirus.**

| Family | Species | Tobamovirus resistance | symptoms | | DAS-ELISA |
| | | | local | systemic | PMMoV |
|---|---|---|---|---|---|
| *Solanaceae* | *Capsicum annuum* "Westlandse Grote Zoete" | | CL, R(2)* | ~C (2)* | 0.371 (local), 2.156 (systemic) |
| | *Capsicum annuum* "CGN19226" | L1 | CL, NL, R (4) | - | 0.109 |
| | *Capsicum frutescens* "CGN21546" | L2 | CL, NL, R (4) | - | 0.240 |
| | *Capsicum chinense* "CGN21557" | L3 | NL (4) | - | 0.102 |
| | *Capsicum chacoense* "CGN21477" | L4 | LA, NL, NR (4) | - | 0.165 |
| | *Solanum lycopersicum* "Moneymaker" | | - | - | 2.118 |
| | *Solanum lycopersicum* "CGN24212" | Tm1 | - | - | 1.975 |
| | *Solanum lycopersicum* "CGN24213" | Tm2 | - | - | 2.131 |
| | *Solanum lycopersicum* "CGN24214" | Tm2$^2$ | - | - | 2.047 |
| | *Solanum lycopersicum* "CGN24218" | Tm1, Tm2/Tm2$^2$ | - | - | 0.084 |
| | *Solanum lycopersicum* "Arbason" | Tm1, Tm2/Tm2$^2$ | - | - | 0.109 |
| | *Solanum lycopersicum* "Cindel" | Tm1, Tm2/Tm2$^2$ | - | - | 0.096 |
| | *Solanum lycopersicum* "Halay" | Tm1, Tm2/Tm2$^2$ | - | - | 0.069 |

Per species four plants were inoculated; the number of plants showing specific symptoms is indicated in brackets. Average $OD_{405}$ values after 2 hours of substrate incubation were considered positive (red) when $\geq$ 2x the value of mock-inoculated controls, otherwise values were considered negative (green). -: no symptoms, ~: mild symptoms, C: chlorosis, CL: chlorotic lesions, LA: leaf abscission, NL: necrotic lesions, NR: necrotic rings, R: rugosity

*two plants showed local symptoms only, two other plants showed systemic symptoms only, DAS-ELISA was performed separately on pooled samples of young leaves of the two plants with local symptoms and systemic symptoms, respectively.

*Solanum torvum* from Thailand (5 samples). The novel virus was not detected in any of these samples.

## Discussion

This study assessed the potential risks of a newly discovered tobamovirus, tentatively named "African eggplant-associated virus" (AEaV), from *S. macrocarpon*. The virus was easily transmitted mechanically, and showed the ability to infect economically important crops, such as eggplant, pepper, petunia, tomato, and other species including *Ammi majus*, *Nicotiana hesperis*, *N. occidentalis* and *Solanum nigrum*. Overall, it was able to infect representatives of two out of six tested plant families.

Sequence analysis indicated that the novel virus had a genome organization characteristic for tobamoviruses, with less than 75% nucleotide identity to other tobamoviruses, which is significantly lower than the tobamovirus species demarcation threshold of 90% nucleotide identity [18]. Therefore, it should be considered as a novel species within the genus *Tobamovirus*. Two genotypes of the novel virus were present in the intercepted *S. macrocarpon* fruit. Although passages in *N. occidentalis* "P1" of the novel virus did not change the nucleotide sequence of either genotype, a decrease in relative frequency of genotype 1 was observed which led to genotype 2 taking over completely in a subsequent passage to *S. macrocarpon*. This might be explained by adaptation to the new host, as fitness cannot be maximized simultaneously in all potential hosts [19, 20].

Remarkably, an NCBI BLAST with the novel virus sequences resulted in a close hit with an alleged bacteriophage isolate, which we expect to be incorrectly assigned and to actually belong to the same novel tobamovirus. This "bacteriophage" sequence was part of a study that analyzed read data from human metagenomic samples [21]. Even though the query cover of the alignment with the novel virus was only 60%, the high nucleotide identity with genotype 1 and

genotype 2, 95.7% and 99.5% respectively, suggests that the same virus was partially present in a human metagenomic sample. Possibly, food or plant material contaminated with the virus had been consumed or touched by a human.

A central paradigm in plant pathology is to demonstrate disease causation from a candidate pathogen based on Koch's postulates. The novel virus was consistently detected in inoculated *S. macrocarpon* plants, but we were not able to replicate the symptoms found on the original fruit. Difficulties in demonstrating causal association are not uncommon in plant virology and the limitations of Koch's postulates have been widely discussed [22]. A potential explanation for the lack of fruit symptoms could be the absence of genotype 1 in the inoculated *S. macrocarpon* plants. Additionally, the development of symptoms after infection is influenced by environmental circumstances and genotype of the plant, or a combination of these factors. Symptom variability as a result of environmental or cultivar effects can be considered as a bottleneck for associating a novel virus with a disease [23]. Alternatively, the diffuse chlorotic mottling observed on the intercepted fruit may have had a non-viral cause and we may have accidentally discovered an asymptomatic infection.

A host range and pathotype study was conducted to assess the potential impact of the novel virus. The infection of multiple solanaceous species was not unexpected, because the novel tobamovirus was discovered in a host from this family. Additionally, *Ammi majus* from the *Apiaceae* family was identified as a host. Although host ranges of tobamoviruses are usually quite narrow in nature, they can be moderate to wide under experimental conditions [18]. Wild species like *Ammi majus* and *S. nigrum* that were infected experimentally might act as virus reservoirs, in which a virus can evolve and spread to other plants and crops [24], for instance to one of the domesticated solanaceous hosts. Such virus reservoirs complicate disease management. Vice versa, infected crops could pose a risk to natural vegetation [25, 26].

For tomato four resistant accessions were identified which all harbored the combination of *Tm1* and *Tm2*/*Tm2²*, whereas accessions with only one of these genes were tolerant, *i.e.* no symptoms were observed while infection was detected by DAS-ELISA. This suggests that a combination of these *Tm* genes is required for resistance. Most commercial tomato cultivars on the European market that are grown in greenhouses harbor this combination of resistances. For open field crops, the use of these resistances is less common, but tobamovirus infections are less likely to spread [27]. Therefore, the impact of the novel virus on tomato cultivation in Europe is likely to be low. It cannot be excluded that another factor in the genetic background of the resistant accessions is involved. Allele dosage might play a role as well, as it is not clear whether the resistances were homozygous or heterozygous in the tested accessions. Even though *Tm2²* has historically been the most durable tobamovirus resistance, it has previously been broken by ToBRFV (5) and partially broken by ToMMV [28]. In a screening of 160 tomato genotypes for ToBRFV resistance, only a single resistant genotype was identified. A locus controlling tolerance mapped to chromosome 11 and was found to interact with a locus on chromosome 2, at the *Tm1* region, to confer resistance [29]. It would be interesting to identify the genetics behind tolerance and resistance to AEaV in tomato as well.

The breaking of *Tm2²* resistance by ToBRFV has been attributed to a replacement of the amino acid cysteine (C68) by histidine (H67) in its movement protein. C68 is conserved among all other *Solanaceae*-infecting tobamoviruses, and is involved in *Tm2²* activation and viral movement [30]. The novel tobamovirus contains C68, but surprisingly also infected a tomato accession harboring *Tm2²*. The breaking of this resistance is therefore not due to the replacement of C68 as was shown for ToBRFV.

For pepper, accessions harboring either of the $L^1$, $L^2$, $L^3$ or $L^4$ genes/alleles were resistant, whereas a cultivar without any of these resistance genes was susceptible. Since most commercial pepper varieties grown in the EPPO region harbor *L* resistance genes/alleles [27], the

impact on pepper in Europe is also likely to be low. However, it cannot be excluded that pepper plants harboring these resistances are susceptible to AEaV under different conditions. For example, an isolate of ToBRFV was able to infect pepper plants harboring $L^{1,3,4}$ at high temperatures, whereas they were otherwise resistant [5].

Due to the advance of high throughput sequencing, novel virus species are being identified at a rapid pace, but biological characterization is often lacking due to resource constraints [23, 31]. However, such experiments are crucial to determine the biological significance and predict the potential impact of a novel virus. The current study identified a novel tobamovirus which experimentally infected multiple plant species, mainly *Solanaceae*, thus being potentially harmful to several economically important crops including eggplant, pepper, petunia and tomato. This study indicated that potato is not a host, while resistance is present in both tomato and pepper germplasm. As most commercial tomato and pepper varieties grown in European greenhouses harbor the relevant resistances, the risk of infection of these crops in Europe is likely to be low. The non-resistant tomato accessions showed no symptoms under the applied experimental conditions, indicating that the economic impact of the novel virus may be low in this crop, even in the absence of resistance. However, the durability of these resistances against the novel virus and their effectiveness under different environmental conditions remain unknown.

In conclusion, the present distribution of the novel virus and its prevalence are unknown. The virus was not detected in a limited survey of solanaceous fruits imported into the Netherlands, which did not include further samples from Mexico. At present, we cannot predict the likelihood of transmission of AEaV to other plant species. In general, hygiene measures and the use of genetic resistance are the main strategies to prevent transmission of tobamoviruses. The observed latent infections in multiple experimental host species underline the merit of screening symptomless plants for viral infections [32]. Therefore, additional monitoring of *S. macrocarpon* and other *Solanaceae* around the world, including non-symptomatic material, is needed to get a better understanding of the incidence, distribution and phytosanitary risk of AEaV and other potentially harmful viruses.

## Supporting information

**S1 Fig. Sequence depth of both virus genotypes in the original sample and after passages in *N. occidentalis* "P1" and *S. macrocarpon*.**
(TIF)

**S2 Fig. Symptoms on pepper accessions (*Capsicum* spp.) inoculated with the novel tobamovirus.** a) *Capsicum annuum* "Westlandse Grote Zoete": chlorotic lesions, b) *C. annuum* "CGN19226": chlorotic lesions, leaf bulging, c) *C. chacoense* "CGN21477": necrotic lesions, leaf abscission, d) *C. chinense* "CGN21557": necrotic lesions, e) *C. frutescens* "CGN21546": chlorotic and necrotic lesions.
(TIF)

**S1 Appendix. Raw DAS-ELISA values of Tables 2–4.**
(XLSX)

## Acknowledgments

We would like to thank Tim Buysman (Kwaliteits-Controle-Bureau, The Netherlands) for intercepting the *Solanum macrocarpon* fruit from which the novel virus was identified. We would also like to thank our colleagues Robert Dees and Joanieke van Oorspronk for performing DAS-ELISA, Jerom van Gemert, Christel de Krom and Carla Oplaat for symptom

observations, Pier de Koning for assistance with Illumina sequencing analysis, Dirk Jan van der Gaag for feedback on an early version of the manuscript, and greenhouse staff for technical support.

## Author Contributions

**Conceptualization:** Anne K. J. Giesbers, Marleen Botermans.

**Data curation:** Anne K. J. Giesbers.

**Formal analysis:** Anne K. J. Giesbers.

**Investigation:** Anne K. J. Giesbers.

**Methodology:** Anne K. J. Giesbers, Marleen Botermans.

**Project administration:** Anne K. J. Giesbers.

**Software:** Marcel Westenberg.

**Supervision:** Annelien Roenhorst.

**Visualization:** Anne K. J. Giesbers, Marcel Westenberg.

**Writing – original draft:** Anne K. J. Giesbers.

**Writing – review & editing:** Anne K. J. Giesbers, Annelien Roenhorst, Martijn F. Schenk, Marcel Westenberg, Marleen Botermans.

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
