## [Decision Letter · Decision Letter 0]

28 Feb 2023

PONE-D-22-30238African eggplant-associated virus: characterization of a novel tobamovirus identified from *Solanum macrocarpon* and assessment of its potential impact on tomato and pepper cropsPLOS ONE

Dear Dr. Roenhorst,

Thank you for submitting your manuscript to PLOS ONE. After careful consideration, we feel that it has merit but does not fully meet PLOS ONE’s publication criteria as it currently stands. Therefore, we invite you to submit a revised version of the manuscript that addresses the points raised during the review process.

ACADEMIC EDITOR: I agree with the comments of the peer reviewer and hence request you to address all the comments carefully and resubmit the manuscript for further consideration.

We look forward to receiving your revised manuscript.

Kind regards,

S.V. Ramesh, PhD

Academic Editor

PLOS ONE

Journal Requirements:

Additional Editor Comments:

The peer-reviewer requires modifications in the current version of manuscript, if authors agree with the comments are requested to submit the revised version for further consideration.

Reviewers' comments:

Reviewer's Responses to Questions

**Comments to the Author**

1. Is the manuscript technically sound, and do the data support the conclusions?

Reviewer #1: Yes

2. Has the statistical analysis been performed appropriately and rigorously? 

Reviewer #1: Yes

3. Have the authors made all data underlying the findings in their manuscript fully available?

Reviewer #1: Yes

4. Is the manuscript presented in an intelligible fashion and written in standard English?

Reviewer #1: Yes

5. Review Comments to the Author

Reviewer #1: In this study, the authors aimed to identify viruses infecting Solanum species and their potential hosts. As a result of RNA sequencing and bioinformatic analyses of imported eggplant fruit from Mexico, a novel virus was identified and named African eggplant-associated virus. This manuscript is valuable for identifying the novel virus and providing information on its potential hosts and resistant cultivars.

The introduction section should focus on known viruses infecting Solanum species and background information on this study.

At the end of the introduction, the purpose of this study, which does not include any results from this study, can be briefly described.

The manuscript is well-written, and the abstract is nicely written.

The results from this study are presented in Figure 1, which could be moved to the Results section.

It is suggested to provide the abbreviation for "The International Plant Protection Convention," and the materials and methods should describe the details on mechanical passages and reagents/equipment's used in this study.

Maximum likelihood method should be used instead of Neighbor-Joining method, and the abbreviation for the virus can be described in Line 168.

Additionally, it would be helpful to generate a new phylogenetic tree for Figure 2 with Maximum likelihood method and to show the genome organization of the novel virus.

The tables show excellent results, and overall, this manuscript is well-done. Congratulations!

6. PLOS authors have the option to publish the peer review history of their article (what does this mean?). If published, this will include your full peer review and any attached files.

Reviewer #1: **Yes: **Won Kyong Cho

---

## [Author Response · Author response to Decision Letter 0]

22 Mar 2023

Dear editor, dear reviewer,

Thank you for the time and effort that you dedicated to provide useful feedback on our manuscript. Please find our response to each point raised by reviewer #1 below:

• Reviewer: In this study, the authors aimed to identify viruses infecting Solanum species and their potential hosts. As a result of RNA sequencing and bioinformatic analyses of imported eggplant fruit from Mexico, a novel virus was identified and named African eggplant-associated virus. This manuscript is valuable for identifying the novel virus and providing information on its potential hosts and resistant cultivars.

Authors: Thank you for the appreciation of our work.

• Reviewer: The introduction section should focus on known viruses infecting Solanum species and background information on this study.

Authors: A very wide range of viruses are known to infect Solanum species, too many to describe in the introduction. The introduction focuses on the Tobamovirus genus, because the novel virus belongs to this group.

• Reviewer: At the end of the introduction, the purpose of this study, which does not include any results from this study, can be briefly described.

Authors: We rephrased the last paragraph of the introduction to clarify the purpose of the study.

• Reviewer: The manuscript is well-written, and the abstract is nicely written.

Authors: Thank you for your appreciation.

• Reviewer: The results from this study are presented in Figure 1, which could be moved to the Results section.

Authors: Figure 1 has been moved to the Results section.

• Reviewer: It is suggested to provide the abbreviation for "The International Plant Protection Convention," and the materials and methods should describe the details on mechanical passages and reagents/equipment's used in this study.

Authors: The abbreviation “IPPC” has been added to line 35. The Materials and Methods section has been slightly restructured, to clarify that the procedure and reagents/equipment of the mechanical passages have been described in Verhoeven and Roenhorst (2000) to which is referred.

• Reviewer: Maximum likelihood method should be used instead of Neighbor-Joining method, and the abbreviation for the virus can be described in Line 168. Additionally, it would be helpful to generate a new phylogenetic tree for Figure 2 with Maximum likelihood method and to show the genome organization of the novel virus.

Authors: We also generated a phylogenetic tree using RAxML (Randomized Axelerated Maximum Likelihood) in Geneious. This has been added to Materials & Methods line 134-135. The positioning of the novel virus within the tree, and the positioning of the other viruses relative to each other, was the same as when the Neighbor-Joining method was used, which has been added to line 177-179. Therefore, Figure 2 presents a reliable phylogenetic tree.

The genome organization of the novel virus corresponds to the genome organization of other tobamoviruses, with four open readings frames (ORFs), encoding a small replicase subunit, an RNA-dependent RNA polymerase (translated through ribosomal readthrough at the stop codon of the small replicase subunit), a movement protein, and a coat protein (as described on line 164-167).

The abbreviation for the virus (AEaV) has been added to line 175 (previously line 168), after which the abbreviation is used.

• Reviewer: The tables show excellent results, and overall, this manuscript is well-done. Congratulations!

Authors: We highly appreciate your feedback and have done our best to improve the manuscript considering your suggestions. 

Yours sincerely,

Anne Giesbers, PhD

---

## [Decision Letter · Decision Letter 1]

30 Mar 2023

African eggplant-associated virus: characterization of a novel tobamovirus identified from *Solanum macrocarpon* and assessment of its potential impact on tomato and pepper crops

PONE-D-22-30238R1

Dear Dr. Roenhorst,

We’re pleased to inform you that your manuscript has been judged scientifically suitable for publication and will be formally accepted for publication once it meets all outstanding technical requirements.

Kind regards,

S.V. Ramesh, PhD

Academic Editor

PLOS ONE

Additional Editor Comments (optional):

Reviewers' comments:

Reviewer's Responses to Questions

**Comments to the Author**

1. If the authors have adequately addressed your comments raised in a previous round of review and you feel that this manuscript is now acceptable for publication, you may indicate that here to bypass the “Comments to the Author” section, enter your conflict of interest statement in the “Confidential to Editor” section, and submit your "Accept" recommendation.

Reviewer #1: All comments have been addressed

2. Is the manuscript technically sound, and do the data support the conclusions?

Reviewer #1: Yes

3. Has the statistical analysis been performed appropriately and rigorously? 

Reviewer #1: Yes

4. Have the authors made all data underlying the findings in their manuscript fully available?

Reviewer #1: Yes

5. Is the manuscript presented in an intelligible fashion and written in standard English?

Reviewer #1: Yes

6. Review Comments to the Author

Reviewer #1: The authors have appropriately addressed the reviewer's comments in their manuscript.

Therefore, I recommend this manuscript for publication as is.

7. PLOS authors have the option to publish the peer review history of their article (what does this mean?). If published, this will include your full peer review and any attached files.

Reviewer #1: No

---

## [Editor Report · Acceptance letter]

5 Apr 2023

PONE-D-22-30238R1 

African eggplant-associated virus: characterization of a novel tobamovirus identified from *Solanum macrocarpon* and assessment of its potential impact on tomato and pepper crops 

Dear Dr. Giesbers:

I'm pleased to inform you that your manuscript has been deemed suitable for publication in PLOS ONE. Congratulations! Your manuscript is now with our production department. 

Kind regards, 

on behalf of

Dr. Shunmugiah Veluchamy Ramesh 

Academic Editor

PLOS ONE